# Nanotechnology as a Tool to Mitigate the Effects of Intestinal Microbiota on Metabolization of Anthocyanins

**DOI:** 10.3390/antiox11030506

**Published:** 2022-03-05

**Authors:** Thiécla Katiane Osvaldt Rosales, Neuza Mariko Aymoto Hassimotto, Franco Maria Lajolo, João Paulo Fabi

**Affiliations:** 1Department of Food Science and Experimental Nutrition, School of Pharmaceutical Sciences, University of São Paulo, São Paulo 05508000, Brazil; thieclarosales@usp.br (T.K.O.R.); aymoto@usp.br (N.M.A.H.); fmlajolo@usp.br (F.M.L.); 2Food and Nutrition Research Center (NAPAN), University of São Paulo, São Paulo 05508080, Brazil; 3Food Research Center (FoRC), CEPID-FAPESP (Research, Innovation and Dissemination Centers, São Paulo Research Foundation), São Paulo 05508080, Brazil

**Keywords:** anthocyanins, antioxidant activity, biotransformation, controlled delivery, intestinal bacteria, metabolism, nanoencapsulation, phenolic compounds, oxidative stress, polysaccharide-based, protein-based, lipid-based

## Abstract

Anthocyanins are an important group of phenolic compounds responsible for pigmentation in several plants. For humans, a regular intake is associated with a reduced risk of several diseases. However, molecular instability reduces the absorption and bioavailability of these compounds. Anthocyanins are degraded by external factors such as the presence of light, oxygen, temperature, and changes in pH ranges. In addition, the digestion process contributes to chemical degradation, mainly through the action of intestinal microbiota. The intestinal microbiota has a fundamental role in the biotransformation and metabolization of several dietary compounds, thus modifying the chemical structure, including anthocyanins. This biotransformation leads to low absorption of intact anthocyanins, and consequently, low bioavailability of these antioxidant compounds. Several studies have been conducted to seek alternatives to improve stability and protect against intestinal microbiota degradation. This comprehensive review aims to discuss the existing knowledge about the structure of anthocyanins while discussing human absorption, distribution, metabolism, and bioavailability after the oral consumption of anthocyanins. This review will highlight the use of nanotechnology systems to overcome anthocyanin biotransformation by the intestinal microbiota, pointing out the safety and effectiveness of nanostructures to maintain molecular stability.

## 1. Introduction

In recent years, nanotechnology has been considered an important tool for the smart delivery of bioactive compounds in the human body. Nanoencapsulation can be an alternative for the accurate release of phenolic compounds, such as anthocyanins, in the human intestine, thus preserving some biological beneficial effects. Nanoencapsulated anthocyanins can be protected from several factors related to human digestion, mainly biotransformation caused by intestinal microbiota while improving absorption [1,2].

Anthocyanins are of great interest because of their many biological activities. Anthocyanins are soluble vegetable pigments from the class of flavonoids [3,4]. The strict connection between anthocyanin and intestinal microbiota has been studied for many years. Regular consumption of anthocyanins can promote intestinal homeostasis, stimulating the growth of beneficial bacteria, thus improving human health [5,6]. On the other hand, intestinal bacteria have a fundamental role in the metabolization of anthocyanins, leading to structural degradation and biotransformation [5,7,8] and also to the production of bioactive metabolites in a reciprocal interaction [9]. The change in anthocyanin molecular structure reduces the absorption and the possible beneficial effects of intact molecules [9,10], but anthocyanins’ metabolites can also be beneficial to humans.

To minimize the extensive degradation of the aromatic ring structures of anthocyanins by microbiota and protect from other factors responsible for biotransformation in the gastrointestinal tract, researchers have been developed viable alternatives to overcome this massive loss [11,12,13]. Polysaccharides, proteins, and lipids are indicated as potential nanocarriers for anthocyanin-loaded systems [14,15,16,17]. Nanoencapsulated anthocyanins can be protected and have controlled release, increasing the absorption of compounds in their integral form with an improvement in bioavailability and antioxidant activity in specific target tissues [18,19,20]. Therefore, this comprehensive review provides knowledge about the role of intestinal microbiota in extensive metabolization and the relation with other diverse health benefits of anthocyanins. Furthermore, the review contains a wide discussion on the possible use of nanotechnology to minimize the effects of microbiota action on anthocyanins and to improve controlled intestinal delivery.

## 2. Anthocyanins and Human Health: Regular Consumption and Associated Benefits

Epidemiological, clinical, and nutritional studies support the evidence of the relationship between the intake of determined classes of food and human health. Studies point to ingestion benefits of fruits and vegetables, since the consumption of these classes of foods has been associated with a reduction in the risk of developing noncommunicable diseases [20,21,22]. In addition, the benefits for human health are related to the ingestion of polyphenolic compounds, such as anthocyanins, as well as some other plant-derived compounds [23,24,25,26,27].

Anthocyanins are water-soluble compounds that are responsible for pigmentation in several plants. Anthocyanins are one of the major subclasses of flavonoids, a class of polyphenols [28]. These phenolic compounds are derived from secondary plant metabolism, mainly distributed in the vacuoles that are inside cell walls (leaves, flowers, and fruits) providing a wide spectrum of colors, such as blue, red, and purple [29]. The color spectrum is directly affected by changes in pH. In acidic conditions anthocyanins have a red color, and when the pH increases, they turn into blue color. Food sources with high anthocyanin content are blackberries, blueberries, strawberries, grapes, and some tropical fruits [30].

It is widely described in the literature that the consumption of anthocyanin-rich foods is associated with various positive effects on human health [6,20,31]. The functional dietary properties are associated with the inhibition of oxidative stress due to potent antioxidant activity and some other metabolic regulations. Thus, when these compounds are ingested regularly, they can contribute to a reduction in the risk of several diseases whose genesis is oxidative stress with further metabolic impairments [24,27].

The main biological effect observed for anthocyanins is the effective antioxidant capacity [3,28,29,30]. They can easily donate protons to highly reactive free radicals, preventing propagation and further radical formation. These compounds are considered excellent antioxidants due to several characteristics. They have a positive charge, aromatic hydroxyl groups in ideal numbers and organization, a fair degree of structural conjugation, and the presence of electron-donor and electron-withdrawn substituents in the ring structure. All these features break the cycle of the generation of new radicals due to electron deficiency [32,33,34,35]. The main mechanisms involved in the biological activity of anthocyanins are related to the free-radical-neutralization pathway, the cyclooxygenase pathway, the protein-kinase pathway, and the signaling of inflammatory cytokines. Anthocyanins can interrupt lipid-oxidation reactions through radical scavenging or as metal chelators to convert metal hydroperoxides or pro-oxidants to stable compounds [32,33]. The outstanding antioxidant capacity of anthocyanins is observed in several in vitro studies. Anthocyanins can neutralize free radicals by donating an electron or hydrogen atom to an extensive range of reactive oxygen species (ROS), such as superoxide (O_2_^−^), singlet oxygen (^1^O_2_), peroxide (RCOO^•^), hydrogen peroxide (H_2_O_2_), hydroxyl radical (OH·), hypochlorous acid (HOCl^−^), peroxynitric acid (ONOOH^−^), and reactive nitrogen species in a terminator reaction [36,37,38,39,40].

The neutralization of radicals by anthocyanins protects cells from oxidative damage, decreasing the risks of aging and various diseases. In this context, many in vitro and in vivo studies confirmed the health benefits attributed to anthocyanins, such as their antioxidant role [30,37,38,39,40,41,42,43,44], anti-inflammatory action [45], neuroprotection [46,47,48,49], anticancer effects [20,46,47,48,49,50,51,52,53,54,55,56], antiobesity effects [21,57,58,59,60,61,62,63,64], cardiovascular protection [65,66,67], antidiabetic effects [68,69,70,71,72,73], visual protection [74,75,76], and antimicrobial properties [72,73,77,78]. A recent systematic review of 44 randomized controlled trials and 15 prospective studies relating to cardiovascular diseases and ingestion of anthocyanin-rich foods or pure anthocyanins showed strong evidence of their effect on improving the blood lipid profile and decreasing circulating proinflammatory cytokines, justifying their inclusion in a cardioprotective diet [73].

Particularly important is the potential effect of anthocyanins on brain health. They have shown neuro-anti-inflammatory properties and promising protection against neurodegeneration diseases associated with aging [79]. In this respect, particularly challenging is the passage of the blood–brain barrier and delivering these compounds to the brain. Some anthocyanins can cross the barrier while others cannot, and in this case, their effect is not in loco but indirect and due to improvement of local circulation [80].

Recently, the antioxidant and anti-inflammatory activities of anthocyanins from *Lycium ruthenicum* Murray were evaluated in animal models after long-term ingestion. The analyzed results indicate that the antioxidant status in the liver was increased and the inflammatory status in the colon was decreased, with a beneficial modulation of intestinal microbiota. Moreover, researchers noticed an increase in short-chain fatty acids in the cecal content and feces. These results are important to prove the long-term effects of anthocyanin intake and support the idea that enriching foods with anthocyanins is effective in modulating intestinal microbiota [24]. The modulation of microbiota is especially relevant to aging [81].

Another important application of anthocyanins is in the food industry. Due to its coloring capacity and water solubility, it allows for easy incorporation into aqueous food systems [82]. Anthocyanin-rich extracts are becoming increasingly attractive for use as a natural substitute for synthetic dyes in the food and pharmaceutical industries, which is an excellent ecologically sustainable alternative [38]. The use of anthocyanins can benefit the sensory quality of food products. Furthermore, the outstanding antioxidant capacity (decrease lipid and protein oxidation) of these flavonoids stimulates several approaches to enable wide technological applicability in the food industry [83,84,85,86,87]. Despite the relationship between health and anthocyanin consumption being evident, and their use in the food industry increasing as natural colorants or even as dietary supplements, the biological characteristics of anthocyanins are directly related to the preservation of molecular stability [33,77].

## 3. Anthocyanin: Chemical Structure and Molecular Stability

Structurally, anthocyanins are in glycosylated form, the basic structure is constituted by an anthocyanidin nucleus linked to sugars and organic acids [75,78]. Anthocyanins possess two benzene rings linked by a linear three-carbon chain. Anthocyanins are soluble in polar solvents (methanol, ethanol, and water). Acidified methanol (stabilization of the flavylium cation) is widely used for extraction [87,88,89].

More than 635 anthocyanins (six common aglycones and various types of glycosylation and acylation forms) have been identified in nature [29]. Because free anthocyanins are unstable, they are mostly found in glycoside form (galactose, rhamnose, arabinose, xylose, and glucuronic acid are the most common) [89]. In addition, some organic acids can be found attached to the hydroxyl groups on the nucleus and/or to the glycosyl units of anthocyanins [90]. Six major glycoside compounds are found in nature, based on the variation of hydroxylation and methoxylation on aromatic rings and also based on the number and positions of the substituents: pelargonidin, delphinidin, cyanidin, peonidin, petunidin, and malvidin [91]. Cyanidin-3-*O*-glucoside and Malvidin-3-*O*-glucoside are the predominant anthocyanins in plants, especially in fruits. They has a positive charge on the C-ring oxygen atom of the basic flavonoid structure [87]. These compounds have structural variations, such as the position and number of hydroxyl in the molecule, the degree of methylation, and the nature and number of the linked sugar molecule [79,83]. Figure 1 shows the molecular structure of anthocyanidins (aglycone form).

However, the color and molecular stability of these pigments are influenced by various factors, such as molecular structure, pH changes, exposure to light, proteins and metallic ions, enzymatic action, and intestinal microbiota [92]. The chemical structure of these compounds, mainly the number and position of the hydroxyl group (-OH) and methoxy groups (-OCH3), influences the molecular stability [25,84]. Furthermore, the pH has a significant influence on the structure and color of the anthocyanins. The variations in pH result in different molecular balances, in which at low pH (acidic) the anthocyanins are particularly red and predominantly in the form of flavylium cation; at slightly acidic pH the structure is colorless hemiketal; and hemiketal chalcone is converted in chalcone by a ring-opening with a yellowish coloration, which at basic pH predominates the quinoidal structure and purple/blue coloration (Figure 2) [93]. The presence of glycosides increases water solubility; on the other hand, acetylation provides higher stability to the anthocyanin molecule [94]. Other factors such as high temperature, processing, storage, and the presence of oxygen also affect stability [95].

Isolated anthocyanins are highly unstable and susceptible to chemical degradation [96]; thus, the measure of human bioavailability and the incorporation into food products are significant challenges [97]. Moreover, molecular instability restricts the use of natural colorants in food systems for processing, formulation, and storage conditions [98]. Thus, due to coloring properties and the numerous health benefits, researchers are involved in exploring the natural potential of anthocyanins. They are interested in developing approaches to maintain molecular structure during food processing and storage through identifying viable alternatives to protect the molecule during digestion, mainly to mitigate the action of the intestinal microbiota.

## 4. Anthocyanin Biotransformation by Human Intestinal Microbiota

In recent decades, research has been directed towards elucidating the complex relationship between anthocyanin consumption and the role of intestinal microbiota. Evidence indicates that long-term consumption of anthocyanins can positively influence human health through positive modulation of intestinal microbiota [23,91]. In addition, microbiota interferes in fundamental biological functions such as absorptive events. The intestinal microbiota is made up of more than a trillion microorganisms established in symbiosis with the host. The systemic effects of the microbiota include immune defense, maintenance of the intestinal barrier, and decreased colonization of potentially pathogenic microorganisms [99,100,101]. Intestinal dysbiosis can impair the bioavailability of numerous essential and nonessential food components. A balanced microbiota provides an increase in intestinal villi and may reduce the risk of developing diseases such as cancer [7,94,95].

Anthocyanins when consumed regularly in foods or supplements can modify the composition of intestinal microbiota, mainly bacteria. In vitro and in vivo studies indicate that certain bacteria with pathogenic potential can have their growth inhibited. On the other hand, the metabolization of anthocyanins by the microbiota can benefit the growth of beneficial bacteria [57,96,97,98]. The main effects of anthocyanins on the intestinal microbiota are related to changes in the composition of bacteria, favoring the specific improvement of the intestinal microbiota population, such as an increase in Bacteroidetes and a decrease in Firmicutes [4]. The imbalance in the bacterial population was observed in animal models induced to obesity (fat diet) but supplemented with high doses of anthocyanins. A reduction in the proportion between the number of Firmicutes and fecal Bacteroidetes was observed, indicating that supplementation with anthocyanins could modulate the animal’s microbiota, thus favoring the reversal of obesity [99,100]. Some factors influence the metabolism of anthocyanins by the intestinal microbiota. The daily ingested dose, the structure of anthocyanin, and interindividual differences have direct interference in the composition of the microbiota [101,102].

The metabolism of anthocyanins by intestinal bacteria involves a sequence of chemical cleavages, initially the glycosidic bonds and then the breaking of the anthocyanidin heterocycle and the degradation to phloroglucinol derivatives and benzoic acids [103,104,105,106,107,108]. Absorption of intact anthocyanins is limited [94], and they are degraded by the action of α-rhamnosidase and β-glycosidase, which are needed to catalyze the reaction, releasing sugar moieties from the anthocyanin structure and transforming it into aglycone form (anthocyanidin) [109,110,111,112,113]. Several intestinal bacteria can metabolize anthocyanins, including *Bifidobacterium* spp. and *Lactobacillus* spp., and the consequent metabolites can stimulate the growth of other specific bacteria, thus providing further modulation of the intestinal microbiota [7,10,113,114,115,116].

Other important factors resulting from the metabolism of anthocyanins by the microbiota are related to short-chain fatty-acid production. Acetate, propionate, and butyrate can serve as a substrate for intestinal epithelial-cell growth (favoring nutrient absorption), can decrease the intestinal pH, and also inhibit the growth of pathogenic bacteria [6,7,106]. Furthermore, anthocyanin supplementation can stimulate an increased number of goblet cells and tight junction proteins and improve villi in the intestine [6].

For the metabolites (low molecular weight) derived from the metabolization of anthocyanins, beneficial effects on the health of the host are attributed [23,107,108], such as the formation of protocatechuic and gallic acids that inhibit the growth of pathogenic bacteria [109,110]. Furthermore, a study using *Wistar* rats supplemented for a long period with high doses of cyanidin-3-*O*-glucoside evaluated the effects on the microbiota after exposure to 3-chloro-1,2-propanediol. The study suggests that anthocyanins contributed to the maintenance of a balanced intestinal microbiota in rats. The evaluated anthocyanin proved to be effective in protecting the intestinal mucosa against damage and in stimulating the growth of beneficial bacteria, restricting intestinal dysbiosis [8]. In this sense, some studies using animal models with oral supplementation of anthocyanins (extracted from different sources and with different concentrations) showed that these compounds influenced the composition of intestinal bacteria in a beneficial way [117,118,119,120,121,122,123]. Thus, the health-promoting effects attributed to anthocyanins are associated with the modulation of the intestinal microbiota [7]. However, despite the several positive effects of anthocyanins described in the literature, there is no consensus on doses and time of ingestion because of intestinal-microbiota variability between humans. This knowledge gap indicates the need for more studies related to the establishment of tolerable upper-intake levels and other dietary guidelines for the consumption and supplementation of anthocyanins [39].

In the elderly, the composition of microbiota changes and may lead to a reduction in concentration and diversity of beneficial bacteria, leading to dysbiosis. Interaction of anthocyanins with microbiota that generates health effects is particularly important for the prevention of diseases in the aging population, with minimal side effects that may occur with drugs [9,81].

### Absorption and Metabolism of Anthocyanins

The metabolism of anthocyanins is a complex process that involves various organs and tissues. In the human host, anthocyanins (from different food sources) undergo successive degradation steps by the action of enzymes and intestinal bacteria, as already described. In addition, the intestinal pH could account for the molecular instability of anthocyanins but could also favor the intestinal biotransformation. Within enterocytes, colonocytes, and in the liver, anthocyanins are metabolized in phase I (less frequently) and phase II [124]. The metabolites generated by the breakdown of the anthocyanin structure and endogenous chemical modifications are excreted via biliary secretion, feces, and urine [118].

Anthocyanins can cross the stomach (pH 1.5 to 2) in their intact form. In vitro digestion simulation studies have found that anthocyanins are generally stable during incubation with gastric fluids [125,126,127,128,129,130,131,132]. In addition, some studies suggest that there is also absorption in the stomach mucosa, due to the rapid detection of anthocyanin absorption markers in the bloodstream after ingestion of food rich in this compound [104,121,122]. However, most of the absorption occurs in the intestine. In the small intestine (pH 7.4–8), mainly in the jejunum, the absorption of glycosylated forms occurs. Anthocyanidins are passively absorbed after the action of hydrolytic enzymes (changing anthocyanins to the aglycone form) and/or the absorption of glycosylated forms occurs through glucose transporters (SGLT1 and GLUT2) [123,124]. Moreover, the integrity of the intestinal villi is critical for absorption [133]. In enterocytes, anthocyanins undergo phase 2 reactions of metabolism, such as methylation, glucuronidation, and sulfation, catalyzed by UDP-glucuronosyltransferase, sulfotransferases, and catechol-*O*-methyltransferases, respectively [10,125]. Figure 3 illustrates the process of anthocyanin metabolism in the human body from the stomach to the excretion of metabolites.

Unabsorbed anthocyanins reach the colon and are metabolized by the colonic microbiota. Most of the absorption of metabolites occurs in the large intestine (pH 7.4–8) [5]. A portion of unabsorbed metabolites and unabsorbed anthocyanins are excreted in feces. A study conducted with patients with ileostomies indicated that most of the anthocyanins arrive in the large intestine intact to be degraded by the microbiota [134]. Additionally, the hydrolysis, reductions, dihydroxylation, demethylation, decarboxylation, and ring fission reactions occur in the colon [127,128]. Bacterial metabolism occurs initially by cleavage of glycosidic bonds, breaking the anthocyanidin heterocycle (C-ring), and degradation to phloroglucinol derivatives (A-ring) and benzoic acids (B-ring) [111]. Figure 4 demonstrates the metabolism of anthocyanins (cyanidin-*O*-glucoside) in the presence of human intestinal bacteria. The degradation process is the result of some conversion steps that are catalyzed by bacterial enzymes in the host. Intestinal bacteria initiate this process by deglycosylation, and then other compounds are formed, such as cyanidin (aglycone), petunidin (a methylation product), and low-molecular-weight catabolites, such as phenolic acids and other phenols. The phenolic acids can then be absorbed by active or passive absorption in the colon and undergo phase 2 enzymatic metabolism [135,136,137].

If absorbed, anthocyanidins and their microbial catabolites are transported through the portal vein and in the liver are distributed to hepatocytes, where they are again metabolized (phases I and II). The products of hepatic metabolism are distributed throughout the tissues and subsequently transported to the enteric system by the bile pathway (an important vehicle for transport) and removed via urinary and/or fecal excretion [35,128].

The absorption of anthocyanins isolated in mixtures or in nanostructured systems is considered a complex mechanism and is not fully elucidated. Anthocyanins may interact differently at diverse absorption sites along the gastrointestinal tract. Advanced techniques are being applied to understand the absorption of these compounds with greater precision, to relate the structure of anthocyanins (isolated or encapsulated) with the absorption and the effect on certain groups of bacteria in the intestinal microbiota. In situ matrix-assisted laser desorption/ionization mass spectrometry imaging can be useful to know the specific sites of absorption and to release anthocyanins (and their metabolites) in different target tissues [138,139,140,141].

The major human metabolites identified in the bloodstream were gallic, vanillic, protocatechuic, 3,4-dihydroxybenzoic, syringic, *p*-cumaric, vanillic, 2,4-dihydroxybenzoic, 2,4,6-trihydroxy benzoic, and 2,4,6-trihydroxy benzoic acids [4,128]. The aglycone form can also be metabolized by intestinal bacteria as a carbon source, decomposing into organic acids such as 3,4-dihydroxyphenylacetic, m-hydroxyphenyl acetic, and *m*-homovanilic acids [131]. However, after ingestion of the anthocyanin source, a limited quantity of intact anthocyanins was detected in the systemic circulation [84,130].

## 5. Biotransformation of Anthocyanins and the Consequent Effect on Bioavailability and Antioxidant Capacity

The metabolism of anthocyanins is complex, and the intense degradation of these compounds limits the bioavailability and the systemic effect. The bioavailability of anthocyanins refers to the amount that are absorbed, reach circulation, suffer metabolization, and are distributed to target tissues [142]. The bioavailability of anthocyanins is very low compared to other flavonoids. In addition to limited absorption and inefficient transport to circulation and distribution, these compounds have high excretion [143]. The biotransformation of anthocyanins by the action of the microbiota leads to less absorption, low biological use, and influences the antioxidant capacity and biological action in specific tissues (39). Inefficient absorption has been reported in some studies, which report that less than 1% of ingested anthocyanins reach the intestine intact and are internalized by enterocytes. Most reports are related to the absorption of metabolites resulting from the degradation of these compounds [102,132,144,145]. Therefore, the low absorption and limited bioavailability of free anthocyanins are due to their susceptibility to high chemical and microbial degradation and excretion rates [133].

The interaction between anthocyanins and the microbiota, and the consequent low bioavailability, has been described in several studies [4,112,113,145,146]. An in vitro study using rat feces evaluated the impact of intestinal bacteria on the degradation of cyanidin-*O*-3-glucoside. The results indicated that anthocyanins were rapidly degraded, which confirms the impact of bacterial action on molecular stability [124]. Some in vivo studies have shown maximum plasma levels of total anthocyanins being 1–100 nM after ingestion of doses at 7–1618 mmol [86,134,135]. After 4 h of ingesting a natural source of anthocyanins, the estimated loss is 60 to 90% that are not detectable in the gastrointestinal tract [94]. In this sense, many in vivo studies have already identified a low absorption and high degradation of anthocyanins by animals and humans [115,147,148,149], probably due to inherent chemical structure but also involving other factors such as food matrix, interaction with nutrients, food processing, and individual factors (genetic and physiological) among other factors [7].

Thus, all mechanisms involved in anthocyanin degradation are still being elucidated. However, the biological activity of the intestinal microbiota is considered an important factor [4,5,6]. Despite the increasing number of studies indicating the possible physiological role of anthocyanin metabolites, greater absorption of anthocyanins (integral form) could increase the antioxidant capacity in specific tissues. In this regard, many researchers are seeking to identify ways to mitigate the effect of microbiota on the biotransformation of anthocyanins [2,11,138,139].

## 6. Nanotechnology Overcoming the Metabolization of Anthocyanins: Biopolymers Delivering Strategies

One of the viable and effective alternatives to minimize the effects of microbiota in the extensive metabolization of anthocyanins is the use of nanotechnology. Nanotechnology is defined as the design, use, and manipulation of materials in systems at the nanometric scale (˂1000 µm) [150,151,152]. Nanocarriers can protect anthocyanin from unfavorable environmental conditions, e.g., pH, temperature, enzyme action, and microbiota degradation [2]. Resistant materials are used to coat the nanostructures, which in addition to protecting anthocyanins during digestion can release them in a controlled manner in the intestine and/or in target cells [11,13]. Furthermore, the anthocyanins encapsulated in the nanostructure could have less interaction with other compounds in the diet, improving bioavailability [2].

A study demonstrated that nanoencapsulated anthocyanin had a greater tolerance to the increase in pH range, the presence of metal ions, and the increase in temperature, thus maintaining the intrinsic capacity of scavenging free radicals [153]. The use of encapsulated anthocyanins, mainly for the formation of biopolyelectrolyte complexes, has shown to maintain stability, overcome chemical degradation, and mitigate the loss of color, thus preserving the bioactivity and enabling their application in foods as natural dyes [154]. The possibility, steps, and strategies were clearly shown in a recent example related to the microencapsulation of polyphenols from *Sambucus nigra* L. [155]. Targeting the intestine is important to control local inflammatory diseases, and recent research designated gut-delivery polyphenols encapsulated with marine polysaccharides as multifunctional nanocarriers [156].

One of the specific chemical properties of anthocyanins refers to their ability to non-covalently interact with some macromolecules to form stable nanostructures [157]. The application of nanocarrier systems for anthocyanin loading can make use of natural polymers, such as polysaccharides, proteins, and lipids [151]. They are pointed out as promising for use as a wall material because they have wide sources of extraction in nature and show excellent biodegradability and biocompatibility [18]. Anthocyanins within the nanostructure are protected from the excessive degradation that happens within the intestinal microbiota. The nanostructures with encapsulated anthocyanins can represent greater absorption of intact molecules by the intestinal mucosa than when free anthocyanins are administered, providing a probable better systemic activity when compared to isolated ingestion [158].

Nanostructures based on polysaccharides can protect and release the encapsulated compounds according to specific physiological stimulation and environment. The physical and chemical properties and functional performance of polysaccharides confer numerous advantages for anthocyanin encapsulation. The complexity of polysaccharide structures is suitable for the construction of nanocarriers. Polysaccharides such as chitosan, cellulose and derivatives, and pectin are widely used for this purpose, protecting and controlling the release of encapsulated bioactive compounds, including anthocyanins [143,144]. Polysaccharide-based nanomaterials are designed for enhancing the responsive delivery that depends on pH, protecting the encapsulated from the intestine environment, and delivering specifically to lower portions of the human intestine. The controlled intestinal release of nanostructures containing anthocyanins can favor absorption, especially in its integral form [153,159,160,161]. The absorption of anthocyanins within polysaccharide nanostructures can occur by recognition of the glycosidic portions of pectin by intestinal epithelial cells in the nanostructures, internalized by the plasma membrane through endosomes, and then the anthocyanins being released in the cell cytoplasm [147,148].

Different types of carbohydrates—natural or modified polysaccharides—are used alone or in combination with other macromolecules to create nanocarriers for anthocyanins delivery [162,163,164,165,166,167,168,169,170,171,172]. Polysaccharide-based nanoencapsulation is suitable for protection, stability, and bioavailability in nanoencapsulation. Over the years, many studies demonstrated the success in the utilization of various polysaccharides for encapsulation of anthocyanins (extract from different sources), such as pectin [164,165,166], chitosan [145,146,152,153,154,155,156,157], and cellulose [173].

The interaction between anthocyanins (cyanidin-3-*O*-glucoside) and citrus pectin with different esterification patterns was investigated using thoroughly explored analytical techniques (isothermal titration calorimetry, nuclear magnetic resonance, and UV-Visible spectrophotometry). The study showed interactions between anthocyanin and pectin, depending on the degree of polysaccharide esterification. It was also reported that a combination of these two compounds had an impact on color maintenance and anthocyanin stability [174]. Furthermore, polysaccharides can form gels when hydrated, have the highest swelling ability, and are ionizable in certain pH ranges, which favors the controlled release and the ability to adhere to animal mucus which improves the delivery to certain organs/tissues [160,161]. In addition, some polysaccharides can be slowly fermented by human intestinal bacteria to an energetic substrate, which will release the encapsulated anthocyanin gradually, thus mitigating molecular degradation that occurs in the intestinal environment [172,175,176,177,178]. Polysaccharides can also interact with proteins forming stable nanostructures at variable pH, therefore protecting encapsulated anthocyanins [162,163].

Proteins are biopolymers extensively used to fabricate nanostructures for the encapsulation of bioactive molecules. Proteins (animal or plant origin) alone or in combination with polysaccharides or other protein molecules can be efficiently used for anthocyanin nanoencapsulation. The protein (natural or denatured form) disposed in the nanostructure provides greater stability to the whole nanocomplex [179]. Examples of proteins used for nanoencapsulation of bioactive compounds are β-Lactoglobulin [1], lysozyme [164], and whey protein [166]. The interaction between anthocyanins at different concentrations (from black rice) and isolated soybean protein was analyzed using three-dimensional fluorescence and Fourier transform infrared spectroscopy. It was observed that anthocyanins are linked by covalent and noncovalent interactions to proteins, with the anthocyanin-protein nanocomplex formed showing a promising use in food formulations [180]. Additionally, the effect of proteins on the stability and bioaccessibility of anthocyanins was recently evaluated. Bovine serum albumin at different concentrations was added to anthocyanins extracted from blueberries in a simulated digestion system and simulated different food processing and storage. The stability and antioxidant capacity of anthocyanins were maintained with the addition of protein, specifically at 0.15 mg/mL. This fact indicated that there was a possible inhibition of anthocyanin degradation by added proteins, thus maintaining the antioxidant capacity [181].

Lipids are also considered suitable as nanocarriers for anthocyanin encapsulation. Lipid-based nanoencapsulation can provide high encapsulation efficiency, controlled release in the intestine, low toxicity, and the excellent possibility of production on an industrial scale. Lipid-based nanoencapsulation can be formed by bilayer structures (usually spherical) with specific polar lipids dispersed in aqueous phases [166,167]. Lipid-based carriers include nanoemulsions, nanoliposomes, solid-lipid nanoparticles, and novel generation of an encapsulation system, namely the nanostructured lipid carrier [182,183,184,185,186]. Some studies using lipids as nanocarriers were successful in maintaining the stability of anthocyanins and preserving the anthocyanin’s chemical structures in a diverse environment [169,170].

In general, biopolymers (polysaccharides, proteins, and lipids) can be applied to the optimization of encapsulation systems. They can be modified or used in their natural form, combined or isolated, and built by different techniques to create smart delivery systems. Biopolymers have potential advantages, such as excellent physicochemical properties, capacity, and functionalities for anthocyanin stabilization techniques [171,172]. However, the nanoencapsulation techniques and the derived nanocompounds should be thoroughly studied by in vitro and in vivo approaches. This is because nanoencapsulated anthocyanins could not perform an ideal antioxidant capacity as the nonencapsulated compounds or could even be degraded if overheated during fabrication; they may even not be released in the target tissue [166]. In this way, researchers must explore the simulated digestion and anthocyanin release in distinct biological systems. Nanoencapsulated anthocyanins may also decrease the effect of food matrices on their absorption [97]. Therefore, incorporating anthocyanins into different food systems is challenging, and nanoencapsulation can be a viable and effective option. It is possible to add them to foods, supplements, and dietetics products [97,187,188]. This could also be a form of increasing the use of underexploited regional fruits and residues from the food industry to develop new products with added economical value, and to explore the existing biodiversity sustainably. Various techniques have been reported in designing nanocarriers based on polysaccharides, proteins, and lipids applied in nanoencapsulation anthocyanins; some of these studies are shown in Table 1. 

In addition, microencapsulation can also be used to stabilize anthocyanins. This encapsulation technology is widely studied to provide greater molecular stability, preserve the antioxidant activity, improve bioaccessibility, and confer controlled-release properties to anthocyanins. Microencapsulation is a process in which the bioactive compound is coated with a specific material to protect against adverse environmental conditions—such as food storage—and intrinsic factors of human digestion [189,190,191,192,193]. In general, microencapsulation refers to the elaboration of a particle with a diameter from 1–1000 µm. There are several types of materials used to microencapsulate anthocyanins, as well as a wide variety of methods for microencapsulation, depending on the purpose of the application, the availability of equipment, and other factors [151,155,189,193].

The methods for elaborating microencapsulated systems can be physical (lyophilization, spray drying, freeze drying, electrospinning/electrospraying), chemical (inclusion complexes), or a combination of both (emulsification, liposomal systems, ionic gelation, and coacervation) [193]. The main biopolymers that can be used as encapsulants are polysaccharides, such as starch, chitosan, pectin, natural gums, mucilage, cellulose, and its derivatives [194,195,196]. Proteins such as whey, caseinate, gelatin, and soy protein are widely used [151,193,197]. The microencapsulation of anthocyanins can be an effective method for the stability, maintenance of color, and antioxidant activity, and has potential for industrial application in foods [151,193,198].

**Table 1 antioxidants-11-00506-t001:** In vitro studies of nanoencapsulation of anthocyanins (polysaccharides, proteins, and lipid-based) for different purposes.

Source	Nanoencapsulant	Nanoencapsulation Technique	Average Size (nm)	Purpose	Reference
Commercial anthocyanin-rich extract	Whey Protein Isolate and Pectin	Thermal processing and electrostatic complexation	200	Increase antioxidant capacity	[166]
Red cabbage	Palmitic acid and surfactants	Emulsion	455	Stability and antioxidant capacity	[199]
Black rice bran	Chitosan and Alginate	Ionic pre-gelation and polyelectrolyte complex	219.53	Stability	[170]
Blueberry	Carboxymethyl Chitosan	Self-assembly	219.53	Protection and stability	[171]
Açai berry	Eudragit^®^ L100	Modified double-emulsion solvent extraction/evaporation	570–620	Safety	[173]
Blueberry	Chitosan Hydrochloride, Carboxymethyl Chitosan	Electrostatic interaction	178.1	Stability and bioavailability	[169]
Blueberry	Whey Protein, Polyglycerol Polyricinoleate	Nanoemulsion	˂400	Protection and stability	[185]
Natural Source Plant	Lecithin and Cholesterol	Nanoliposomal	53.01	Stability and bioavailability	[186]
Blueberry	Chitosan Hydrochloride, Carboxymethyl Chitosan, and β-Lactoglobulin	Electrostatic interaction	91.71	Stability and bioavailability	[168]
Black rice	Chitosan/Chondroitin sulfate	Self-assembly	350.1	Antioxidant capacity	[200]
Red raspberry pomace	β-Lactoglobulin	Desolvation	129.13–351.85	Stability and bioavailability	[1]
Bilberry	Chitosan and Pectin	Self-assembly	100–300	Stability and bioavailability	[172]
Black carrot	Chitosan	Ionic gelation	274	Increase antioxidant capacity	[201]
Blackberry Commercial anthocyanin-rich extract	Pectin and Lysozyme Casein and Carboxymethyl Cellulose	Self-assembly Self-assembly	198.5 209.9	Protection and stability Stability	[164] [202]

## 7. Conclusions and Future Trends

Anthocyanins have a wide spectrum of biological activities, such as antioxidant, anti-inflammatory, or chemopreventive features, which support human health, although their low bioavailability and extensive biotransformation interfere with these advantages. Studies point to the promising application of nanotechnology tools to encapsulate anthocyanins, thus representing a beneficial alternative to maintain molecular stability. Although the studies were successful in the nanoencapsulation of anthocyanins, in vivo studies (animal and human) are still an unexplored field of research. Several studies indicate the promising application of nanoencapsulation anthocyanins in foods, favoring stability during food processing and storage, preservation of sensory characteristics, resistance to environmental conditions, and digestion factors. Future research could focus on the development of fortified foods and nutritional supplements with nanoencapsulated anthocyanins, increasing the supply of food products beneficial to human health. Although all evidence supports the biological beneficial effects of anthocyanin nanoencapsulation, further studies are needed to determine values limits for safe intake. Natural biopolymers demonstrated adequate biocompatibility, biodegradability, and efficiency for anthocyanin delivery and increased bioavailability. Nanoencapsulation based on polysaccharides, proteins, and lipids can protect anthocyanins through the gastrointestinal tract, releasing them in a controlled manner. The use of nanotechnology for smart protection, controlled delivery, and tissue-specific delivery can minimize the effects of microbiota on the biotransformation of anthocyanins, which represents more effective absorption of intact forms and preservation of biological effects, such as antioxidant activity and some other metabolic modulations.

## Figures and Tables

**Figure 1 antioxidants-11-00506-f001:**
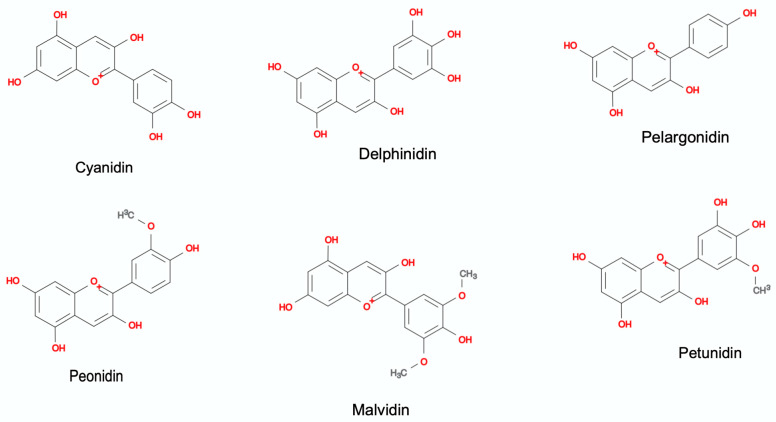
Molecular structure of anthocyanidins (Cyanidin, Delphinidin, Pelargonidin, Peonidin, Malvidin, and Petunidin). The figure was created with Mind the Graph (https://mindthegraph.com (accessed on 10 February 2022)).

**Figure 2 antioxidants-11-00506-f002:**
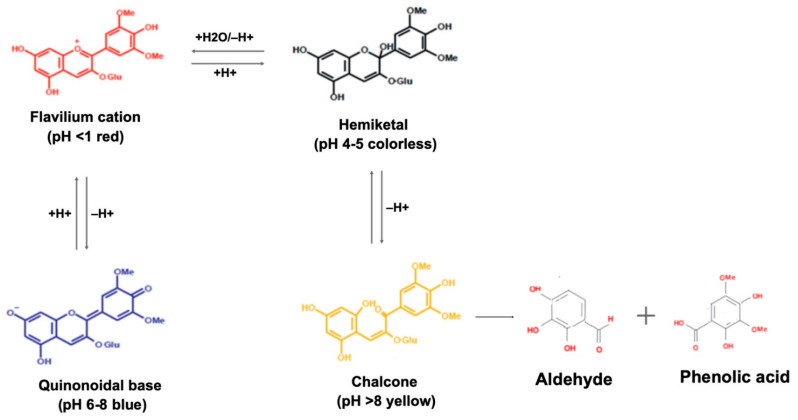
Structural transformation of anthocyanidins at acidic to neutral conditions. The figure was created with Mind the Graph (https://mindthegraph.com (accessed on 10 February 2022)).

**Figure 3 antioxidants-11-00506-f003:**
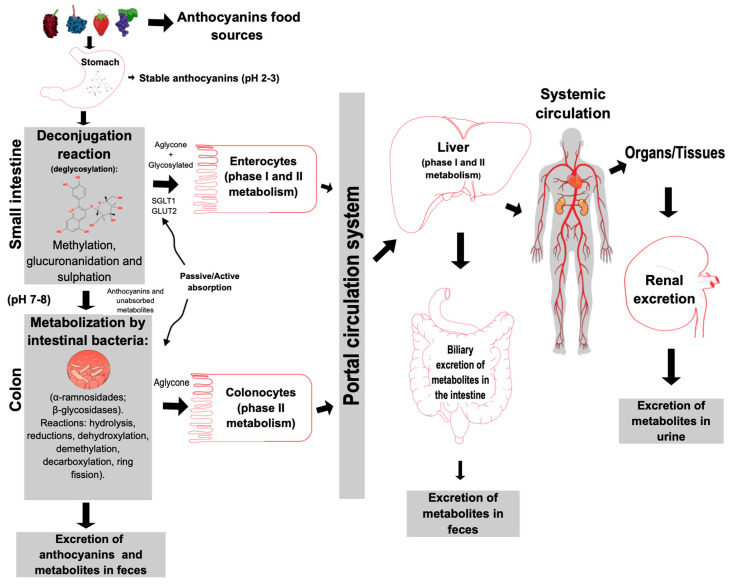
Destination of anthocyanins and their metabolites in the human body after ingestion from food sources. The extensive and successive degradation of anthocyanins by the action of intestinal bacteria and enzymes and the formation of metabolites. After absorption, different organs and tissues are responsible for the metabolization in phases I and II and the excretion of their metabolites. The figure was created with Mind the Graph (https://mindthegraph.com (accessed on 10 February 2022)).

**Figure 4 antioxidants-11-00506-f004:**
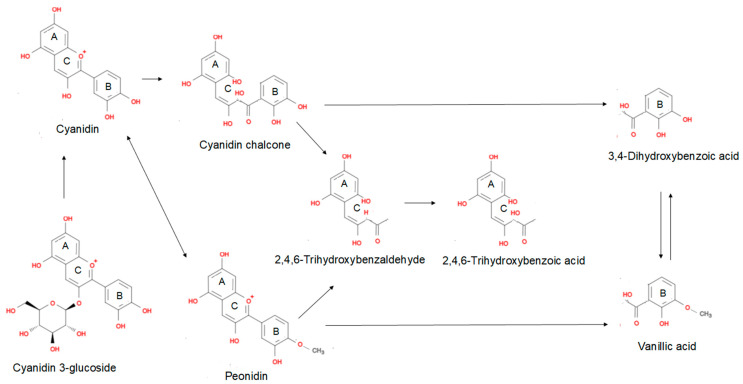
Anthocyanin metabolism by the intestinal microbiota and the formation of different acids. Based on [124,138,139]. The figure was modified from Mind the Graph (https://mindthegraph.com (accessed on 10 February 2022)).

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
