# Peer review of "Nanotechnology as a Tool to Mitigate the Effects of Intestinal Microbiota on Metabolization of Anthocyanins"

_antioxidants, 2022, doi:10.3390/antiox11030506_

Round 1

Reviewer 1 Report

Dear Author,

It is very interesting and well conducted.  only some minro typo error must be revised.

Author Response

Article Reference: #1613757

Journal: Antioxidants

Title: Nanotechnology as a Tool to Mitigate the Effects of Intestinal Microbiota on Metabolization of Anthocyanins

RESPONSE TO REVIEWERS

The authors thank the reviewers and the editor for their precise and valuable suggestions. We have modified the manuscript accordingly and have responded to all criticisms raised. We hope that now you will find it suitable for publication.

Reviewers' comments:

REV#1: It is very interesting and well conducted.  only some minro typo error must be revised.

Answer: We thank the reviewer for the suggestions. The manuscript was revised through a web-based English company to improve expressions and spelling, such as those pointed out by the reviewer.

Reviewer 2 Report

Before I read this paper I was fairly unaware of the nanoencapsulation field as a way to get more dietary anthocyanins through the gut lining in an intact manner.  It seems like a worthwhile field of study.  Probably the idea is about 15 years old, so having a review paper is probably worth while.  However, I have a question whether encapsulation of the anthocyanins so that they pass through the stomach and the intestine unabsorbed, is actually better?  Maybe getting more absorbed in the colon when encapsulated is not overall so great?  Maybe, in fact, the total amount per ingestion is worse overall because the stomach and intestine didn't contribute ?  From what I read there's no in vivo data to answer this, so it is probably unknown.  But, I think this could also be true, and so I propose it could be discussed. 

A few other things:

It is incorrect from my scan of the literature to say there’s a consensus that the intestinal microbiota is the preponderant factor in how much anthocyanin is absorbed.

“…all mechanisms involved in anthocyanins degradation are still being elucidated. However, it is a consensus that the biological activity of the intestinal microbiota is the preponderant factor (94).”

The above quote runs in the face of reference #94 which says nothing about a consensus.  (94) Prior RL, Wu X. Anthocyanins: structural characteristics that result in unique metabolic patterns and biological activities. Free Radic Res. 2006 Oct;40(10):1014-28. 

--------------------------------

Not all previous studies agree that encapsulation is helpful.  I think it needs more balance.  For instance, here’s the conclusion of one previous study:

“Overall, our results show that the encapsulation of anthocyanins within the biopolymer particles fabricated in this study was not particularly effective at improving their antioxidant activity or color stability. Alternative strategies are therefore needed to encapsulate this important color and nutraceutical agent.”

This was taken from reference #163.  (163)  Arroyo-Maya, I. J., & Mcclements, D. J. (2015). Biopolymer nanoparticles as potential delivery systems for anthocyanins: Fabrication and properties. Food Research International, 69, 1–8

--------------------------

A few incomplete references exist.

  1. Ayala-fuentes JC, Chavez-santoscoy RA. Nanotechnology as a Key to Enhance the Benefits and Improve the Bioavailability 532 of Flavonoids in the Food Industry. 2021;
  2. Tie S, Tan M. Current Advances in Multifunctional Nanocarriers Based on Marine Polysaccharides for Colon Delivery of 819 Food Polyphenols. J Agric Food Chem. 2022;
  3. Cui H, Si X, Tian J, Lang Y, Gao N, Tan H, et al. Anthocyanins-loaded nanocomplexes comprising casein and carboxymethyl 904 cellulose: stability, antioxidant capacity, and bioaccessibility. Food Hydrocoll. 2022;122(July 2021).

--------------------------------

Your reference list might add the following new paper that shows that just adding an extra protein (in the right dietary amount) can help with absorption of anthocyanins; and if this is the case then going to the length of protein encapsulation, per se, might be over doing it.

Zang Z, Chou S, Si X, Cui H, Tan H, Ding Y, Liu Z, Wang H, Lang Y, Tang S, Li B, Tian J. Effect of bovine serum albumin on the stability and antioxidant activity of blueberry anthocyanins during processing and in vitro simulated digestion. Food Chem. 2022 Mar 30;373(Pt B):131496. doi:

“BSA at different concentrations, specifically at 0.15 mg/mL, inhibited the degradation of ANs and the loss of antioxidant capacity.”

Author Response

The authors thank the reviewers and the editor for their precise and valuable suggestions. We have modified the manuscript accordingly and have responded to all criticisms raised. We hope that now you will find it suitable for publication.

REV#2: Before I read this paper I was fairly unaware of the nanoencapsulation field as a way to get more dietary anthocyanins through the gut lining in an intact manner.  It seems like a worthwhile field of study.  Probably the idea is about 15 years old, so having a review paper is probably worth while.  However, I have a question whether encapsulation of the anthocyanins so that they pass through the stomach and the intestine unabsorbed, is actually better?  Maybe getting more absorbed in the colon when encapsulated is not overall so great?  Maybe, in fact, the total amount per ingestion is worse overall because the stomach and intestine didn't contribute ?  From what I read there's no in vivo data to answer this, so it is probably unknown.  But, I think this could also be true, and so I propose it could be discussed. 

Answer: We thank the reviewer for the precise reading of the article and the suggestions. We did some adjustments in the text to address the reviewer’s concerns.

The reviewer is right when pointing out anthocyanins are absorbed intact in the stomach and intestine. However, one should understand anthocyanins are chemically unstable molecules (impairing the use as dietary supplement or additive) and the absorption occurs in low grade in the stomach (lines 272-276). Moreover, anthocyanins are not fully absorbed in the small intestine (lines 276-286 - Figure 3). New techniques could be applied to understand specifically the sites of absorption and release of anthocyanins (and their metabolites) in different target tissues (lines 321-329). Thus, reminiscent anthocyanins are biotransformed by the colon microbiota (lines 295-299). Both the absorption of intact molecules and microbiota-derived metabolites are beneficial to humans, and we made it clear in the text (lines 46-48, 238-241). However, the absorption is, overall, not complete as demonstrated by the excretion of anthocyanins in feces (lines 295-299). So nanoencapsulation is a promising technique to maintain molecular stability of anthocyanins in food matrices (lines 388-396, 444-454, 516-517) and/or to increase the absorption of intact anthocyanins (lines 404-407, 417-422, 530-534). Furthermore, nanoencapsulation could deliver anthocyanins to specific organs, such as the colon, mitigating massive biotransformation by intestinal microbiota and promoting health benefits locally and systemically (lines 58-60, 393-396, 414-422, 436-439). This would be the differential of the concept, and we fully explored this concept in the text.

We trust that hop we have addressed the reviewer’s concern and the new version could reach the expectation to be considered for publication.

A few other things:

It is incorrect from my scan of the literature to say there’s a consensus that the intestinal microbiota is the preponderant factor in how much anthocyanin is absorbed.

“…all mechanisms involved in anthocyanins degradation are still being elucidated. However, it is a consensus that the biological activity of the intestinal microbiota is the preponderant factor (94).”

The above quote runs in the face of reference #94 which says nothing about a consensus.  (94) Prior RL, Wu X. Anthocyanins: structural characteristics that result in unique metabolic patterns and biological activities. Free Radic Res. 2006 Oct;40(10):1014-28. 

Answer: We understand the reviewer’s point, so the text was modified (lines 367-373).

--------------------------------

Not all previous studies agree that encapsulation is helpful.  I think it needs more balance.  For instance, here’s the conclusion of one previous study:

“Overall, our results show that the encapsulation of anthocyanins within the biopolymer particles fabricated in this study was not particularly effective at improving their antioxidant activity or color stability. Alternative strategies are therefore needed to encapsulate this important color and nutraceutical agent.”

This was taken from reference #163.  (163)  Arroyo-Maya, I. J., & Mcclements, D. J. (2015). Biopolymer nanoparticles as potential delivery systems for anthocyanins: Fabrication and properties. Food Research International, 69, 1–8

Answer: In this study, the authors report that they observed an increase in the stability of anthocyanins to heat. However, encapsulated anthocyanins had a lower antioxidant activity than non-encapsulated anthocyanin during storage, which was attributed to the thermal processing step required for the elaboration of the nanoparticles and the binding of anthocyanins to the biopolymers within the nanoparticles. Perhaps after the release of anthocyanins from the nanoparticles into the target tissue, the antioxidant activity might be preserved, but it was not evaluated. In this way, a consideration was introduced in the text (lines 475-480).

--------------------------

A few incomplete references exist.

  1. Ayala-fuentes JC, Chavez-santoscoy RA. Nanotechnology as a Key to Enhance the Benefits and Improve the Bioavailability 532 of Flavonoids in the Food Industry. 2021;
  2. Tie S, Tan M. Current Advances in Multifunctional Nanocarriers Based on Marine Polysaccharides for Colon Delivery of 819 Food Polyphenols. J Agric Food Chem. 2022;
  3. Cui H, Si X, Tian J, Lang Y, Gao N, Tan H, et al. Anthocyanins-loaded nanocomplexes comprising casein and carboxymethyl 904 cellulose: stability, antioxidant capacity, and bioaccessibility. Food Hydrocoll. 2022;122(July 2021).

Answer: The references were updated.

--------------------------------

Your reference list might add the following new paper that shows that just adding an extra protein (in the right dietary amount) can help with absorption of anthocyanins; and if this is the case then going to the length of protein encapsulation, per se, might be over doing it.

Zang Z, Chou S, Si X, Cui H, Tan H, Ding Y, Liu Z, Wang H, Lang Y, Tang S, Li B, Tian J. Effect of bovine serum albumin on the stability and antioxidant activity of blueberry anthocyanins during processing and in vitro simulated digestion. Food Chem. 2022 Mar 30;373(Pt B):131496. doi:

“BSA at different concentrations, specifically at 0.15 mg/mL, inhibited the degradation of ANs and the loss of antioxidant capacity.”

Answer: We thank the reviewer to point out this issue. The reference was included in the text and some considerations were incorporated into the manuscript (lines 454-461).

Reviewer 3 Report

 These review outlined the nanotechnology techniques for anthocyanin metabolization. Generally is well written but some informations still nedeed. 

I suggest to insert in the chapter 6  microencapsulation methods techniques  to achieve stable microencapsulated formulations (more details such as: lyophilization, spray-drying, ionic gelation or other methods used). 

Author Response

The authors thank the reviewers and the editor for their precise and valuable suggestions. We have modified the manuscript accordingly and have responded to all criticisms raised. We hope that now you will find it suitable for publication.

REV#3: These review outlined the nanotechnology techniques for anthocyanin metabolization. Generally is well written but some informations still nedeed. 

I suggest to insert in the chapter 6  microencapsulation methods techniques  to achieve stable microencapsulated formulations (more details such as: lyophilization, spray-drying, ionic gelation or other methods used). 

Answer: We thank the reviewer for the suggestions. The microencapsulation was not the main focus of the work since the possible passive absorption - as it can occur with some nanocompounds through incorporation into intestinal epithelial cells - will not occur. However, we understand this is an important issue, so we included some ideas about microencapsulation in the text as suggested (lines 490-508).

Round 2

Reviewer 2 Report

This is a more balanced review paper now.  Some run-on sentences might be made to read more simply, but otherwise I like it. 

Reviewer 3 Report

Dear Authors,

The review looks better now!